# The Exosomes of Stem Cells from Human Exfoliated Deciduous Teeth Suppress Inflammation in Osteoarthritis

**DOI:** 10.3390/ijms25168560

**Published:** 2024-08-06

**Authors:** Chuang-Yu Lin, Parichart Naruphontjirakul, Te-Yang Huang, Yi-Chia Wu, Wei-Hsuan Cheng, Wen-Ta Su

**Affiliations:** 1Department of Biomedical Science and Environmental Biology, Kaohsiung Medical University, Kaohsiung 807378, Taiwan; tina422942@gmail.com; 2Regenerative Medicine and Cell Therapy Research Center, Kaohsiung Medical University, Kaohsiung 807378, Taiwan; smallwei2018@gmail.com; 3Biological Engineering Program, Faculty of Engineering, King Mongkut’s University of Technology Thonburi, Bangkok 10140, Thailand; cherrytp828@gmail.com; 4Department of Orthopedic Surgery, Mackay Memorial Hospital, Taipei 104217, Taiwan; haunht33@gmail.com; 5Division of Plastic Surgery, Department of Surgery, Kaohsiung Medical University Hospital, Kaohsiung 807378, Taiwan; 6Department of Chemical Engineering and Biotechnology, National Taipei University of Technology, Taipei 106344, Taiwan; cf1811@ntut.edu.tw

**Keywords:** stem cells from human exfoliated deciduous teeth (SHED), exosome, osteoarthritis (OA), SW1353 cells, hyaluronic acid

## Abstract

Hyaluronic acid injection is commonly used clinically to slow down the development of osteoarthritis (OA). A newly developed therapeutic method is to implant chondrocytes/stem cells to regenerate cartilage in the body. The curative effect of stem cell therapy has been proven to come from the paracrine of stem cells. In this study, exosomes secreted by stem cells from human exfoliated deciduous teeth (SHED) and hyaluronic acid were used individually to evaluate the therapeutic effect in slowing down OA. SHED was cultured in a serum-free medium for three days, and the supernatant was collected and then centrifuged with a speed difference to obtain exosomes containing CD9 and CD63 markers, with an average particle size of 154.1 nm. SW1353 cells were stimulated with IL-1β to produce the inflammatory characteristics of OA and then treated with 40 μg/mL exosomes and hyaluronic acid individually. The results showed that the exosomes successfully inhibited the pro-inflammatory factors, including TNF-α, IL-6, iNOS, NO, COX-2 and PGE2, induced by IL-1β and the degrading enzyme of the extrachondral matrix (MMP-13). Collagen II and ACAN, the main components of the extrachondral matrix, were also increased by 1.76-fold and 2.98-fold, respectively, after treatment, which were similar to that of the normal joints. The effect can be attributed to the partial mediation of SHED exosomes to the NF-κB pathway, and the ability of exosomes to inhibit OA is found not inferior to that of hyaluronic acid.

## 1. Introduction

Cartilage is the smooth tissue that covers the ends of bones in joints, consisting of collagen, proteoglycans and other proteins, embedded in a small number of cartilage cells and is the only tissue not vascularized [1,2]. Chondrocytes are the only cells found in the cartilage tissue and are mainly responsible for regulating the synthesis and degradation of the extracellular matrix (ECM). The main components of the ECM are proteoglycans and collagen. Collagen is the most abundant component in the ECM, accounting for 60% of cartilage weight, which has a stabilizing effect on the matrix. Among all types of collagen, Collagen II accounts for 95%. The second most abundant macromolecule in the ECM is the proteoglycan, with Aggrecan (ACAN) accounting for most of its composition. Proteoglycan contains many chondroitin sulfate and keratan sulfate chains, which form glycosaminoglycan chains that allow the ECM to contain a large amount of water. It, in turn, causes expansion and pressure to produce compressive strength, ultimately resisting cartilage deformation and compression [3]. 

Osteoarthritis (OA) is the most common form of arthritis. It occurs most frequently in the cartilage lesions within a joint. OA can cause pain, stiffness and swelling, and in some cases, reduced function and disability that severely affect the ability of people in handling daily tasks. OA is primarily caused by multiple risk factors, of which aging and obesity are the most prominent reasons. The current disease control is mainly focused on the relief of symptoms by self-management, exercise and weight loss as relevant [4,5]. Previous studies showed that a reduction in the ECM due to chondrocyte damage is the major cause of OA, which eventually leads to cartilage wearing and fibrosis [1]. Healthy cartilage is the buffer to reduce the impact of shock from movement. Patients with OA experience pain, stiffness and loss of motion due to cartilage wearing and tearing and friction between bones. Although the viscosupplementation of hyaluronic acid was used for symptomatic OA treatment, the efficacy of the intra-articular injection of hyaluronic acid remained controversial [6]. 

OA is often accompanied by the destruction of articular cartilage and chronic inflammation, characterized by articular cartilage damage, chondrocyte apoptosis and ECM degradation. These phenomena eventually lead to the deterioration of connective tissues and joints and, ultimately, to osteoarthritis, which results from the presence of various pro-inflammatory cytokines in OA chondrocytes, e.g., interleukin-1β (IL-1β), tumor necrosis factor-α (TNF-α), interleukin-6 (IL-6). Once the joint is being stimulated and those inflammatory cytokines are activated, the chondrocyte will release a matrix metallopeptidase (MMP), a disintegrin and a metalloproteinase with thrombospondin motifs (ADAMTS). The MMP-13, one of the degradation enzymes in the MMP family, will degrade type II collagen, and the ADAMTS5, another enzyme belonging to the ADAMTS family, can degrade proteoglycan, which eventually results in ECM disassembly/imbalance [7]. At the same time, cyclooxygenase (COX) and nitric oxide synthase (NOS) in the cells will also be stimulated by the influence of pro-inflammatory factors. COX includes two categories: COX-1 and COX-2. While COX-1 usually exists in normal cells, COX-2 exists in inflammatory cells as an inducing enzyme that induces the conversion of arachidonic acid into prostaglandin E2 (PGE2) [8]. PGE2 will then stimulate the production of MMP-13 and ADAMTS5, which leads to the degradation of the ECM [9] and finally causes chondrocyte apoptosis [10]. NOS includes neural nitric oxide synthase (nNOS), endothelial nitric oxide synthase (eNOS) and inducible nitric oxide synthase (iNOS), among which iNOS can induce a large amount of NOS under stimulation. NO is an inducing factor for various tissue damage or an enhancing factor for lesion expansion and can induce apoptosis of chondrocytes [11]. The massive production of these pro-inflammatory factors (IL-1β, IL-6, iNOS, NO, COX-2, PGE2) will cause joint inflammation and cytotoxicity of chondrocytes and finally lead to ECM imbalance, apoptosis and symptoms of tissue redness, swelling, burning sensation and pain. 

NF-κB has been recognized as a major key factor involved in the pathological process of OA [12]. In OA chondrocytes, a number of metabolic and chemotactic factors mediated by NF-κB, such as TNF-α, IL-1β, IL-6 and MMP, are expressed, which lead to reduced type II collagen and proteoglycan synthesis and inflammation [13]. In addition, NF-κB induces the production of COX-2, PGE2, iNOS and NO to promote metabolic factor synthesis, cartilage inflammation and apoptosis to enhance joint injury [14].

SW1353 is a chondrosarcoma cell, which is commonly used in arthritis-related experiments. Although the experimental results may be slightly different from those of the primary cartilage, the cultivation process is simpler than that of the primary cartilage and easy to obtain. SW1353 induced by the inflammatory factors used in this study did show inflammatory symptoms. Therefore, we used the SW1353 cell line as an in vitro model of arthritis [15,16,17].

IL-1β is an important pro-inflammatory factor in arthritis. It has been shown to play an important role during arthritis response by severely inhibiting the chondrocyte synthesis factors and inducing catabolic factors, such as the enhanced COX-2 activity, the up-regulated MMP family expression and the increase in NO production [17,18]. The process of cartilage destruction involves multiple pathways. IL-1β is able to produce pro-inflammatory factors and degradative enzymes through activation of the MAPK pathway (ERK, p38, JNK) and the NF-κB pathway [19], which result in OA pathology.

The mesenchymal stem cells (MSCs) from many sources, such as bone marrow and adipose, with clinical potential, provided alternative options for OA treatments. However, due to the efficacy variation and high cost, a consensus of opinions for treatment has not been achieved [20,21,22]. The stem cells from human exfoliated deciduous teeth (SHED) are the tissue-specific stem cells and are one of the MSCs. They reside in the dental pulp cavity of teeth and are an ideal source for stem cells. Many studies have demonstrated the repair function of SHED in wound healing, diabetes mellitus [23] and acute kidney injury [24], as well as anti-inflammatory and anti-fibrotic effects [25]. MSCs can regulate gene expression in the environment with paracrine secretions, which, in turn, can play roles in therapy, protection, inhibition, etc. Paracrine secretions are found in various biological fluids, such as blood, breast milk, urine, etc., in which the external vesicles (EVs) are the important cellular communication mediators secreted by MSCs [26]. Many studies have demonstrated the cartilage-repairing and inflammation-reducing effects of exosomes in OA. Injections of exosomes derived from human embryonic MSCs have been shown to repair the cartilage defects and regenerate cartilage [27]. Exosomes derived from adipose stem cells have been shown to promote cartilage proliferation and inhibit inflammation [28]; exosomes derived from platelet-containing plasma have been shown to promote proliferation and inhibit the apoptosis of chondrocytes [29]. Previous studies have shown that exosomes possess a promising therapeutic modality. The features of small size, good stability, high bioactive contents and specific targeting ability make exosomes a special delivery system with better biocompatibility than liposomes or polymeric nanoparticles. The lipid bilayer structure allows exosomes to be easily absorbed by cells and subsequently enter the cytoplasm in order to deliver their contents [30,31]. In this research, we would like to set up an OA disease model with SW1353 cells, in which the inflammatory signaling pathways can be triggered by IL-1β for studying the anti-inflammatory ability of exosomes from SHED for future OA treatment applications.

## 2. Results

### 2.1. Characterization of Extracted Exosomes from SHED Cultured Medium 

Figure 1 shows the characterization of exosomes harvested from the cultured medium of SHED. Figure 1A shows the morphology of stem cells from SHED by phase contrast microscopy. Figure 1B shows a TEM image of extracted exosomes. Figure 1C shows the molecular markers, e.g., CD63 and CD9, of exosomes and the positive results of Western blotting. In Figure 1D, exosome analysis of the particle size was conducted, and the major size range of exosomes was distributed between 100 and 200 nm in diameter. Figure 1E shows the human chondrosarcoma cells (SW1353).

### 2.2. Cell Viability Test

#### 2.2.1. Exosome Toxicity Tests

To confirm whether the exosome was toxic to the SW1353 cells or not before performing the efficacy test, cytotoxicity tests were executed. The range of exosome dosage is mainly outlined in two studies, which is between 20 μg/mL and 80 μg/mL [32,33]. Since the concentration range is wide, a series of concentrations of 20, 40, 60 and 80 μg/mL were used to treat the cells for two days. As shown in Figure 2A, the survival rates of all concentrations were higher than that of the control group, which indicated that the exosomes secreted by SHED were not cytotoxic to the SW1353 cell line. The survival rate became higher along with the increase in exosome concentration, which indicated that the exosomes might have certain proliferative efficacy on SW1353.

#### 2.2.2. The Effect of IL-1β on the Cell Viability of SW1353 Cells

IL-1β is a key factor in triggering OA. Therefore, we tested the toxic effects with different concentrations of IL-1β (0, 1, 10, 50 and 100 ng/mL) on SW1353 and examined the cell viability at 24 and 48 h. The results in Figure 2B show that as the concentration of IL-1β increased, the survival rate of the SW1353 decreased at both 24 and 48 h and began to show a significant decrease at 10 ng/mL. The survival rate decreased by nearly 50% after two days of treatment with 100 ng/mL of IL-1β, the maximum dose tested, which indicated that the toxicity of IL-1β to the cells was still tolerable and was suitable to induce mild or moderate inflammation. Therefore, a concentration of 10 ng/mL was chosen to treat the SW1353, by which the cell survival rate dropped to 70% after 48 h modelling the inflammatory condition. 

#### 2.2.3. The Effect of Exosomes on the Survival of Inflammatory Cells

After inducing the inflammatory condition with 10 ng/mL IL-1β for 48 h, the SW1353 was then treated with different concentrations of exosomes (0, 20, 40, 60, 80 μg/mL), and the cell viabilities were examined at 24 and 48 h. As shown in Figure 2C, the survival rate did not show much difference among different concentrations. Other than that, cell proliferation was increased at all concentrations of exosomes, and the cell viabilities increased with time, which indicated that the exosomes were capable of inhibiting inflammation and protecting the SW1353 chondrocytes from death. The cells treated with a concentration of 40 μg/mL of exosomes showed the highest survival rate. Therefore, a concentration of 40 μg/mL was chosen for a subsequent 48 h treatment. 

#### 2.2.4. The Effect of Hyaluronic Acid on Inflammatory Cell Survival

To compare the efficacy of SHED exosomes with hyaluronic acid, the cells were treated with IL-1β (10 ng/mL) for 48 h, followed by exosomes (40 μg/mL) and hyaluronic acid (40 μg/mL) individually for another 48 h.

The injection of hyaluronic acid is one of the most common therapeutic OA treatments. According to previous studies, hyaluronic acid does have the effect of inhibiting inflammation [34,35]. In order to compare the anti-inflammation effect with that of exosomes, the same concentration of hyaluronic acid as the one used for exosomes was preferentially used to treat the cells. As the results in Figure 2D show, the survival rates of hyaluronic-acid-treated cells and of exosomes were 92.7% and 98.6%, respectively, both similar to the healthy state of the control group. A concentration of 40 μg/mL of both hyaluronic acid and exosomes, which can potentially protect cartilage and promote cell proliferation, was used for the subsequent experiments.

### 2.3. Inflammatory Markers

#### 2.3.1. The TNF-α and IL-6

TNF-α and IL-6 are both multifunctional cytokines, which are key inflammatory factors in OA [36,37] and are also the most frequently detected pro-inflammatory factors in OA studies [38]. The stimulated TNF-α promotes chondrocyte catabolism and inhibits cell recovery via the NF-κB pathway [39]. The stimulated TNF-α also promotes the release of MMP and results in ECM degradation [40], which is related to the synthesis of other pro-inflammatory factors, such as iNOS and NO. As shown in Figure 3A,B, after stimulation with IL-1β, the TNF-α protein expression increased by 1.6-fold compared to the control group and decreased by 20% after exosome treatment, while it decreased by 70% with hyaluronic acid. Both exosomes and hyaluronic acid showed the recovery effect by inhibiting TNF-α expression. However, the effect of hyaluronic acid is more significant than that of exosomes. Even more, exosome treatment was not able to reverse TNF-α expression to a level lower than that of the control group. 

IL-6 can be induced by IL-1β and TNF-α stimulation [41] and can synergize with TNF-α to inhibit collagen II synthesis and increase MMP production [42]. After treatment with exosomes and hyaluronic acid, the inhibitory effect on IL-6 (Figure 3C,D) was 30% with exosomes so the decreased tendency did not reach the control level and was 65% with hyaluronic acid, which is lower than the control level.

#### 2.3.2. The iNOS and NO

NO is a key inflammatory mediator that plays a critical role in mediating OA [43,44,45] and is induced by iNOS. iNOS can produce large amounts of NO in response to immune or inflammatory stimuli, such as lipopolysaccharides (LPS), IL-1β, TNF-α, etc., which can lead to cartilage destruction. NO upregulates MMP, inhibits ECM synthesis and induces COX-2 and PGE2 production [46]. Nitrite (NO_2_^−^) is the ultimate stabilizer of NO, which is cytotoxic and pro-inflammatory when formed [47]. 

As shown in Figure 4A,B, the iNOS protein expression increased to approximately two-times compared to that of the control group after IL-1β-induced inflammation but was downregulated to a level close to that of the control group after exosomes and hyaluronic acid treatment. As shown in Figure 4C, the NO production after inflammation increased to 6.36-fold of the control group. The effect of inhibiting NO production of exosomes and hyaluronic acid was examined. NO was found to be reduced by 50% by treating either exosomes or hyaluronic acid, which suggested a significant inhibiting effect on NO. The results of iNOS expression showed that exosomes and hyaluronic acid have similar inhibiting effects, and the NO production was reduced as well. Therefore, the NO production mediated by iNOS can be inferred as being interrupted by treating either hyaluronic acid or exosomes. 

#### 2.3.3. COX-2 and PGE2

PGE2 promotes inflammatory mechanisms through various pathways, including the upregulation of MMP [48], enhancement of the ECM degradation [49] and promotion of cartilage apoptosis [10]. COX-2 is the key to induce PGE2, which is triggered by pro-inflammatory factors [50]. Inhibition of COX-2 is one of the mainstays of the current treatment of OA, and non-steroidal inflammatory drugs (NSAIDs) are known as COX inhibitors. As shown in Figure 4D–F, after the treatment of exosomes and hyaluronic acid, the inhibition effects on COX-2 and PGE2 were significant. The exosomes downregulated COX-2 protein expression by 70% and hyaluronic acid by 50%, which were both lower than that of the control group. PGE2 production was reduced by 50% and 40% by exosomes and hyaluronic acid, respectively, which were close to that of the control group. The exosomes were more effective than hyaluronic acid in inhibiting COX-2 and PGE2.

### 2.4. The Degrading Enzyme of Extrachondral Matrix

MMP-13 is the main enzyme for ECM degradation, of which the target is more limited to the connective tissue compared to other MMP family enzymes [51]. MMP-13 degrades not only collagen type II, the main component of the ECM, but also proteoglycans, type IV and type IX collagen [52]. As shown in Figure 5A–C, the protein and the gene expressions of MMP-13 increased 4-fold and 8-fold, respectively, in terms of the inflammatory status. After treatment with exosomes and hyaluronic acid, the protein of MMP-13 was inhibited by 54% and 60%, respectively. The gene expression was almost undetectable, which was close to the level of the control group. The results showed that both exosomes and hyaluronic acid can effectively inhibit MMP-13, which approach the normal joint cartilage status.

The aggrecanases, including the family members ADAMTS1, ADAMTS4, ADAMTS5, ADAMTS8 and ADAMTS9, are responsible for the cleavage of aggrecan in articular cartilage [53,54]. ADAMTS5 is considered the primary degrading enzyme, exhibiting significantly greater proteinase activity compared to other aggrecanases [55]. Figure 5D–F demonstrates a slight increase in both the protein and gene expressions of ADAMTS5 under inflammatory conditions. Following treatment with exosomes and hyaluronic acid, the ADAMTS5 protein was inhibited by 44% and 20%, respectively, while the gene expression was lower than that of the control group. The results showed that exosomes exhibited slightly better efficacy compared to that of hyaluronic acid.

### 2.5. The Extrachondral Matrix

The two major components of the extrachondral matrix are Collagen II and ACAN, and the degradation of the ECM is the main feature of OA. Therefore, promoting the synthesis and inhibition of the metabolism of the ECM has become one of the therapeutic strategies for treating OA. The effective treatment of hyaluronic acid and exosomes can be determined by the increasing amount of the ECM. In this study, the major ECM components, collagen II and ACAN, were examined.

#### Collagen II and ACAN

The main macromolecule of articular cartilage is Collagen II. After a series of stimulations by pro-inflammatory factors, MMP-13 is finally induced to degrade Collagen II, which is shown in Figure 6A–C. Under inflammation status, the protein and gene contents of Collagen II are indeed reduced by 50% and 87%, respectively. The protein was elevated by 1.76-fold and 1.55-fold, respectively, and the gene increased 32.61-fold and 28.77-fold, respectively, after exosome and hyaluronic acid treatment. Exosomes and hyaluronic acid had a similar ability to increase Collagen II levels.

ACAN is the main protein degraded by ADAMTS5. In Figure 6D–F, ACAN protein and gene expressions decreased by 70% and 95.02%, respectively, in the inflammatory state. The protein expression increased by 2.98-fold and 2.72-fold after exosome and hyaluronic acid treatment, respectively. The gene expression also increased significantly.

### 2.6. NF-κB Gene Expression and Activity

It is known that NF-κB is involved in an important pathway for IL-1β to induce SW1353 inflammation by producing various pro-inflammatory factors [56] and degradative enzymes [57]. We first found that the gene expression was increased nearly 2-fold in the inflammatory status (Figure 7A), whereas the exosomes and hyaluronic acid were able to reverse the gene transcripts to the levels of almost not detectable. In combination with the inhibition of the pro-inflammatory factors and the degradative enzymes, it is suggested that the exosomes can inhibit inflammatory symptoms by mediating NF-κB.

When NF-κB is activated, secretory embryonic alkaline phosphatase (SEAP) is secreted [58]. The quantitative SEAP was used as an indicator, and NF-κB activity was indirectly inferred by the SEAP content. The results shown in Figure 7B are similar to the gene expression in Figure 7A. The SHED exosomes and hyaluronic acid were both effective in inhibiting NF-κB.

## 3. Discussion 

In this study, differential centrifugation was used to collect exosomes secreted by SHED. After verifying the morphology, particle size and protein markers in Figure 1B–D, most of the round particles secreted by SHED were confirmed as exosomes containing CD9 and CD63, with a diameter of 119.9 ± 8.1 nm.

According to Figure 2A, the exosomes do not have cytotoxicity to SW1353, and the survival rate of SW1353 is higher as the concentration of exosomes increases. The optimal concentration of the inflammation-inducing drug, IL-1β, was tested. As shown in Figure 2B, IL-1β toxicity toward SW1353 is most prominent at a concentration of 10 ng/mL, and the survival rate of the cells dropped to 70.18% after incubation for two days. The test result of drug concentration, 10 ng/mL, is consistent with that of previous studies [33,59]. Since the increased survival rate of SW1353 after exosome treatment was based on the results of an MTT assay, the effect of total cellular metabolism activity and cell proliferation could not be excluded and requires further verification in the future.

The optimal concentration of exosomes for treating anti-inflammation in the OA model was tested. As shown in Figure 2C, the survival rate reached its highest at a concentration of 40 μg/mL but decreased at higher concentrations of 60 and 80 μg/mL. The ability of exosomes to promote proliferation dominates the survival rate under the condition of no inflammation. In Figure 2D, the ability to inhibit pro-inflammatory factors and degradative enzymes and to promote ECM synthesis is also taken into account in the presence of inflammation. According to the literature, the treatment of exosomes at high concentrations is potentially counterproductive for anti-inflammation [60,61]. Therefore, it was hypothesized that the anti-inflammatory ability decreased at concentrations of 60 and 80 μg/mL without affecting the survival rate. It is possible that the anti-inflammatory effect was not strong or was not at the peak of the assay. 

Abnormally elevated expression of IL-1β, TNF-α and IL-6 in OA patients has been recognized as a key to cartilage damage [39,62]. In this study, the levels of TNF-α and IL-6 proteins were indeed the highest in the inflamed group and were accompanied by the synergistic effect of these three cytokines (IL-1β, TNF-α and IL-6) in stimulating the production of other inducible enzymes (iNOS, COX-2) [63]. The protein expression of iNOS and COX-2 was found to be higher than that of the control group in the stimulated state, and NO and PGE2 also increased along with their respective inducing enzymes, by 6.36-fold and 2.55-fold, respectively, to reach the OA state. Inflammatory (TNF-α, IL-6) and degradation (NO, PGE2) markers were significantly regulated by exosomes and hyaluronic acid treatment. An inflammatory expression protein (TNF-alpha, IL-6) assay showed that hyaluronic acid had a significantly better ability to reduce inflammation than exosomes, and the regulation of iNOS/NO also showed similar results, while COX-2/PGE2 was slightly better in exosomes than in hyaluronic acid. These results indicate that exosomes and hyaluronic acid have the ability to regulate key pro-inflammatory factors in the OA model. This confirms that SHED exosomes have anti-inflammatory ability in the OA model.

The degradation of the ECM is the main feature of OA. The main components of the ECM are Collagen II and ACAN. According to the protein analysis of Collagen II and ACAN, exosomes and hyaluronic acid have the effect of promoting the synthesis of ECM proteins (Figure 6A–D). Recent studies related to the insight of mechanobiology of the dental pulp stem cells (DPSCs) showed that DPSCs with the capability to secrete the ECM can support tissue regeneration and improve osteogenic differentiation [64,65]. That might hint that the exosomes from SHED may be involved in cytoskeleton organization and ECM regeneration. However, to prove this point of view still requires further study.

NF-κB is a cytokine-induced transcription factor that plays an important role in regulating the expression of various genes, including various pro-inflammatory cytokines, adhesion molecules and proteases in arthritis [12]. Therefore, blocking the activity of NF-κB may be an effective treatment for OA. According to the NF-κB gene and activity assay (Figure 7A,B), exosomes and hyaluronic acid can indeed regulate NF-κB expression and have protective and anti-inflammatory effects on SW1353 cells, partly by modulating the NF-κB pathway. This study showed that exosomes from the SHED can inhibit OA of SW1353 induced by IL-1β, as shown in Figure 8; the effect was comparable to that of hyaluronic acid.

### Limitations of the Study

This study showed that the exosomes from SHED can be beneficial for decreasing the inflammatory materials induced by IL-1β in the in vitro SW1353 cell culture system for modelling OA. However, one limitation is that the results can only illustrate the effect of SHED exosomes involved in specific signaling pathways of NF-κB in OA pathological progression; the downstream molecular regulation, such as COX-2 and PGE2, still needs further experiments to verify. The MSC exosomes could be an option for patients, yet they require further studies to set up more comprehensive platforms for evaluation of the repair capacity for OA.

## 4. Materials and Methods

### 4.1. The Stem Cells from Human Exfoliated Deciduous Teeth (SHED) and SW1353 

The SHED was approved by the Animal Ethics Committee and was provided by Kaohsiung Medical University, Kaohsiung, Taiwan (KMUHIRB-SV(I)-20210047). The Human Chondrosarcoma (SW1353) was purchased from the Bioresource Collection and Research Center (Hsinchu, Taiwan). 

The SHED cells were isolated and preserved in our laboratory previously [32] and were then cultured in alpha-MEM (containing 10% Fetal bovine serum and 1% antibiotic-antimycotic; Gibco, Fisher Scientific, Sweden). The culture medium was changed every 2 to 3 days at 37 °C and 5% CO_2_. The cells were passaged when cells were of 80% confluency. SW1353 cells were cultured in DMEM/F12 (containing 10% fetal bovine serum and 1% antibiotic-antimycotic; Gibco, Fisher Scientific, Sweden). The culture medium was changed every 2 to 3 days, and the culture conditions were 37 °C and 5% CO_2_. We passaged the cells when they reached 80% confluency.

### 4.2. Harvest Exosomes

Remove the culture medium and rinse with 37 °C PBS twice. Add 400 μL of trypsin and shake the culture dish to rinse the cells completely. Place in incubator at 37 °C for 5 to 10 min. Confirm that the cells are in suspension under microscope, and add 3 mL of culture medium to neutralize the trypsin effect and rinse the cells. Collect the cells into a 15 mL centrifuge tube and centrifuge at 1000 rpm for 5 min. Remove the supernatant, add 1 mL of fresh culture medium and rinse the cells. Take appropriate amount of cell suspension to 15 cm culture dish and add 14 mL of fresh culture medium. Return to incubator and continue incubation. Replace the medium when the cells in the 15 cm dish are 70% full. Draw out the old medium and lubricate with 37 °C PBS twice. Add serum-free alpha-MEM (containing 1% antibiotic-antimycotic and 1% bovine serum albumin). Return to incubator and collect after three days of incubation.

### 4.3. Exosome Extraction

Collect serum-free culture medium in 50 mL centrifuge tube at 4 °C; centrifuge at 2000× *g* for 10 min to remove dead cells. Take the supernatant and centrifuge at 12,000× *g* for 30 min at 4 °C to remove cell debris and large vesicles. Take the supernatant and filter it through a sterile 0.22 μm filter to remove the remaining impurities. Centrifuge the filtered medium at 100,000× *g* for 70 min at 4 °C. Remove the supernatant, add PBS and centrifuge at 100,000× *g* for 70 min to wash away the remaining protein. Remove the supernatant, add 100 μL of PBS and store at −80 °C for a long time. Before the experiment, use BCA protein to quantify exosomes.

### 4.4. Cell Viability Assay

Seed the cells in a 96-well culture dish (1 × 10^5^ cells/mL) for 24 h. Remove the old medium and add serum-free medium containing different concentrations of IL-1β for 48 h. Remove the old medium and add serum-free medium containing different concentrations of exosomes and process for 48 h. Add 20 μL of MTT solution (5 mg/mL) and incubate at 37 °C for 4 h. Discard the supernatant and add 150 μL of DMSO to each well. Oscillate the 96-well culture dish on a horizontal oscillator for 10 min. Measure the absorbance at 570 nm using a spectrophotometer (Multiskan™ FC, Thermo Scientific™, Waltham, MA, USA).

### 4.5. Western Blotting

The cells were lysed by RIPA buffer with protease inhibitor. The BCA protein assay was performed to determine the protein concentration. After performing the SDS-PAGE, the proteins were transferred from the gel to PVDF membrane (Millipore, Darmstadt, Germany). The gel percentage used for SDS-PAGE depended on the molecular weight of targeted proteins. By following the conventional protocol, the membrane was blocked with 5% skimmed milk, then incubated at 4 °C overnight. After TBST washing, the secondary antibodies conjugated with HRP reacted with the membrane at room temperature for 1 h. To detect the target proteins, the substrates were added. Images were obtained via chemiluminescence image system (LAS-4000, Fujifilm, Santa Clara, CA USA). The antibodies used in this study were as follows: CD63 (1/1000, Abcam, Cambridge, UK), CD9 (1/1000, Abcam, Cambridge, UK), Anti β-actin (1/3000, Abcam, Cambridge, UK), TNF-α (1/1000, Affinity Biosciences, Jiangsu, China), IL-6 (1/1000, Affinity), iNOS (1/1000, Affinity Biosciences, Jiangsu, China), COX-2 (1/1000, Cell signaling technology, MA, USA), MMP-13 (1/1000, Affinity Biosciences, Jiangsu, China), ADAMTS5 (1/1000, Affinity Biosciences, Jiangsu, China), Type II collagen (1/1000, Affinity Biosciences, Jiangsu, China), ACAN (1/500, Affinity Biosciences, Jiangsu, China). 

### 4.6. qPCR

The cells were harvested and lysed to extract the total RNA by the TRIzol (Sigma-Aldrich, St. Louis, MO, USA) reagent. To reverse transcribe the RNA to cDNA, the IScript cDNA synthesis kit (Bio-rad, CA, USA) was used to synthesize the cDNA. For qPCR reaction, the cDNA, primers and SYBR^®^ green (iTaq Universal SYBR^®^ green supermix, Bio-rad, CA, USA) were well mixed. The reaction was detected by the StepOne™ Real-Time PCR System (ThermoFisher Scientific, Waltham, MA, USA). The relative quantification of gene expression was determined by 2^−△△Ct^. The primers used in this study were as follows: MMP-13: F_CCTTGATGCCATTACCAGTCTCC; R_AAACAGCTCCGCATCAACCTGC. NF-κB: F_TGAACCGAAACTCTGGCAGCTG; R_CATCAGCTTGCGAAAAGGAGCC.

### 4.7. Nitrite Detection

Seed cells in a 6-well (1 × 10^6^ cells/mL) plate for 24 h. Remove the old medium and add serum-free medium containing 10 ng/mL IL-1β for 48 h. Remove the old medium and add serum-free medium containing 40 μg/mL exosomes and 40 μg/mL hyaluronic acid for 48 h. Remove the old medium and lubricate with PBS once. Add 1 mL of PBS, scrape the cells with a spatula and collect them into 1.5 mL Eppendorf tube. Centrifuge the cells at 3000 rpm for 5 min at 4 °C. Remove the supernatant and add 100 μL cold assay buffer. Conduct ultrasonic shaking at 100 W for 1 sec at 4 °C and leave standing for 2 s; repeat for 1 min. Centrifuge at 10,000× *g* for 15 min at 4 °C. Remove the supernatant. Add 100 μL of sample, 10 μL of Griess reagent I, 10 μL of Griess reagent II and 80 μL of buffer to 96-well plate in order. Protect from light and shake at room temperature for 10 min. Detect the absorbance at 540 nm via ELISA (Multiskan™ FC, Thermo Scientific™, USA). The absorbance value was taken into the standard curve to obtain the concentration of nitrite.

### 4.8. Determination of NF-κB Activity

Seed cells in a 6-well (1 × 10^6^ cells /mL) plate for 24 h. Remove the old medium and add serum-free medium containing 10 ng/mL IL-1β for 48 h. Remove the old medium, add serum-free medium containing 40 μg/mL exosomes and 40 μg/mL hyaluronic acid, and process for 48 h. Collect the supernatant. Add 180 μL of Quanti-Blue solution and 20 μL of supernatant to 96 wells. Allow to stand for 2 h at 37 °C, protected from light. Detect the absorbance at 650 nm using ELISA (Multiskan™ FC, Thermo Scientific™, USA).

### 4.9. Statistical Analysis

All data were expressed as mean and standard deviation. Statistical analyses were performed using IBM SPSS Statistics 25 software (IBM corp. New York, NY, USA), if applicable, and differences between groups were assessed using one-way ANOVA with the LSD method, with *p* < 0.05 being considered statistically significant.

## Figures and Tables

**Figure 1 ijms-25-08560-f001:**
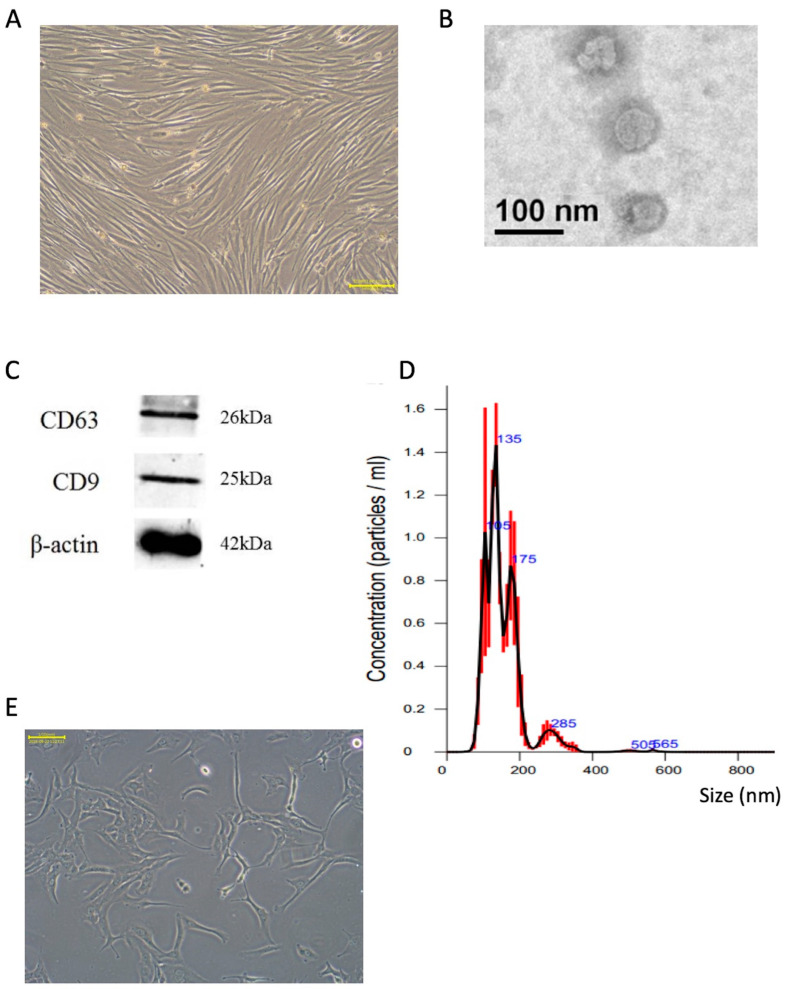
Exosome features. (**A**) The stem cells from human exfoliated deciduous teeth (SHED), scale bar 20 μm. (**B**) TEM image of purified exosome morphology, scale bar 100 nm. (**C**) The molecular markers of exosomes. (**D**) The range of diameters of exosomes. (**E**) Image of cell morphology of the cell SW1353, scale bar 100 μm.

**Figure 2 ijms-25-08560-f002:**
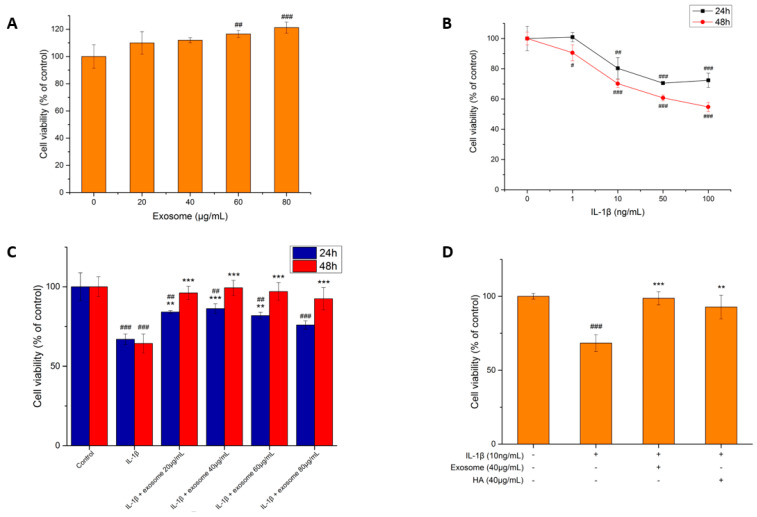
Cytotoxicity tests. (**A**) Cytotoxicity test of exosomes against SW1353 cell line. (**B**) Effect of IL-1β on the survival of SW1353 cell line. (**C**) Effect of exosomes on inflammatory cell survival. (**D**) Comparison of survival rates between exosomes and hyaluronic acid treated for 48 h, respectively. Values are expressed as mean ± SD (n = 3), compared with the control group or the first group **: *p* < 0.01, ***: *p* < 0.001 and #: *p* < 0.05, ##: *p* < 0.01, ###: *p* < 0.001.

**Figure 3 ijms-25-08560-f003:**
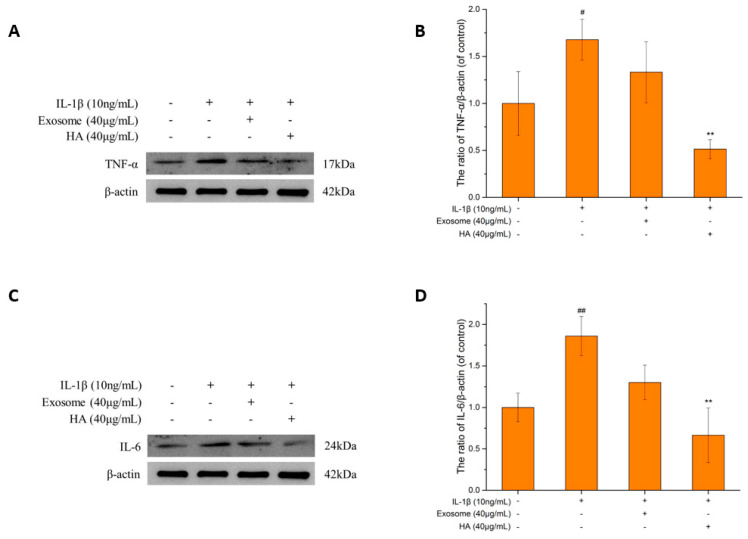
The inflammatory markers TNF-α and IL-6. (**A**) TNF-α protein expression. (**B**) Quantification of TNF-α protein expression. (**C**) IL-6 protein expression. (**D**) Quantification of IL-6 protein expression. Values are expressed as mean ± SD (n = 3), compared with the first group **: *p* <0.01 and compared with the second group #: *p* < 0.05, ##: *p* < 0.01.

**Figure 4 ijms-25-08560-f004:**
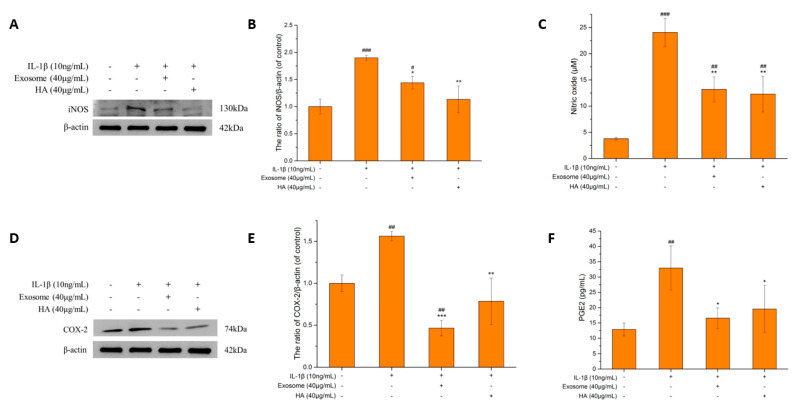
The inflammatory markers iNOS, NO, COX-2 and PGE2. (**A**) iNOS protein expression. (**B**) Quantification of iNOS protein expression. (**C**) Comparison of nitric oxide concentration. (**D**) Quantification of COX-2 protein expression. (**E**) Quantification of COX-2 protein expression. (**F**) Comparison of PGE2 concentration. Values are expressed as mean ± SD (n = 3), compared with first group *: *p* < 0.05, **: *p* < 0.01, ***: *p* < 0.001 and #: *p*< 0.05, ##: *p* < 0.01, ###: *p* < 0.001.

**Figure 5 ijms-25-08560-f005:**
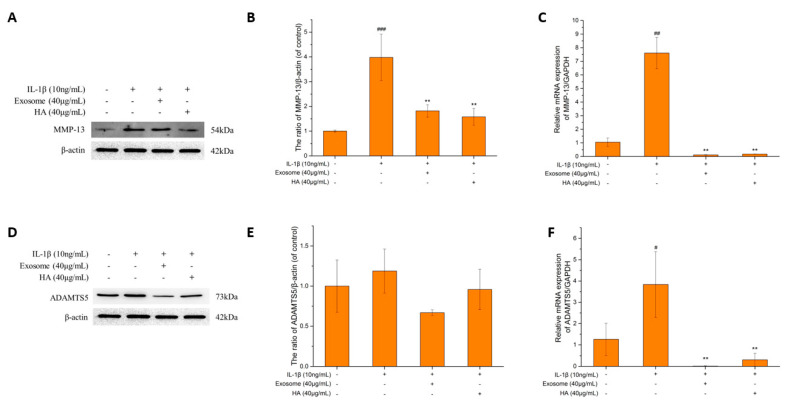
The MMP-13 and ADAMTS5 protein expression. (**A**) MMP-13 protein expression. (**B**) Quantification of MMP-13 protein expression. (**C**) Quantification of MMP-13 mRNA performance. (**D**) ADAMTS5 protein expression. (**E**) Quantification of ADAMTS5 protein expression. (**F**) Quantification of ADAMTS5 mRNA performance. Values are expressed as mean ± SD (n = 3), compared with the first group **: *p* < 0.01 and #: *p*< 0.05, ##: *p* < 0.01, ###: *p* < 0.001.

**Figure 6 ijms-25-08560-f006:**
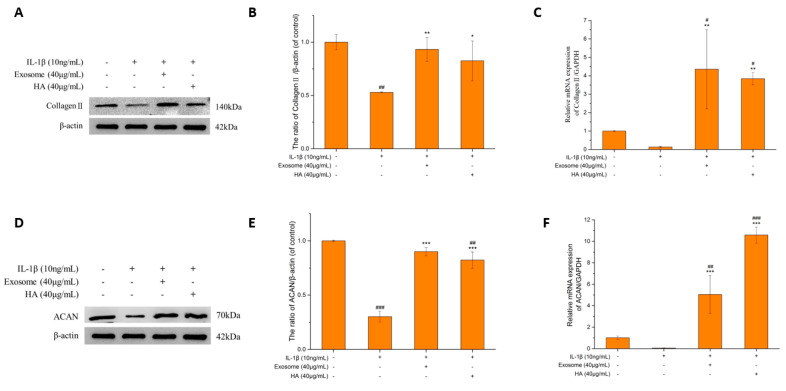
The cartilage extracellular matrix. (**A**) Collagen II protein expression. (**B**) Quantification of Collagen II protein expression. (**C**) Quantification of Collagen II mRNA performance. (**D**) The ACAN protein expression. (**E**) Quantification of ACAN protein performance. (**F**) Quantification of ACAN mRNA performance. Values are expressed as mean ± SD (n = 3), compared with first group *: *p* < 0.05, **: *p* < 0.01, ***: *p* < 0.001 and #: *p*< 0.05, ##: *p* < 0.01, ###: *p* < 0.001.

**Figure 7 ijms-25-08560-f007:**
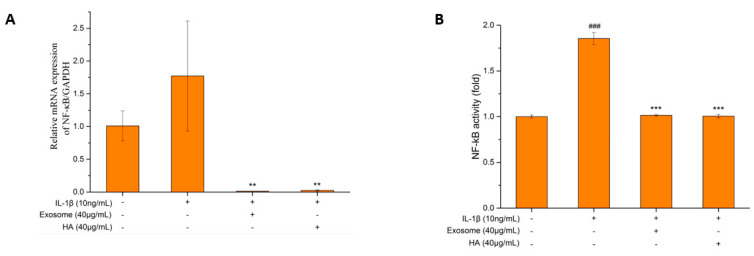
NF-κB gene expression and activity. (**A**) Quantification of NF-κB mRNA performance. (**B**) The activity of NF-κB. Values are expressed as mean ± SD (n = 3), compared with first group **: *p* < 0.01, ***: *p* < 0.001 and ###: *p* < 0.001.

**Figure 8 ijms-25-08560-f008:**
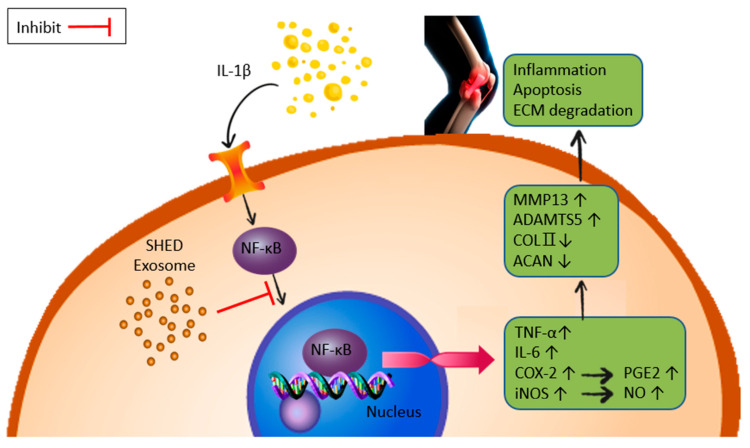
Schematic diagram illustrating the inhibition of the OA pathway by SHED exosomes.

## Data Availability

Data is contained within the article.

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
