# Peer review of "The Exosomes of Stem Cells from Human Exfoliated Deciduous Teeth Suppress Inflammation in Osteoarthritis"

_ijms, 2024, doi:10.3390/ijms25168560_

Round 1

Reviewer 1 Report

Comments and Suggestions for Authors

In this study, exosomes secreted by stem cells from human exfoliated deciduous teeth (SHED) and hyaluronic acid were used to evaluate the therapeutic ability to slow down OA. SHED exosomes were first collected and characterized, and SW1353 cells were stimulated with IL-1β to produce the inflammatory features of OA and then treated with exosomes and hyaluronic acid. Osteoarthritis-related indicators and components of the extracellular matrix were examined, and exosomes were found to have an inhibitory capacity for OA no less than hyaluronic acid. But the paper needs some improvements before acceptance for publication.

1. "ECM" appears in the first paragraph of “introduction” first. The abbreviations should be defined at the first appearance. This should be revised on other abbreviations, too.

2. lt's partly unclear what is the purpose of this paper. Improve this part in the introduction section.

3. The clarity of some figures in the article is low, especially Fig. 1.

4. The authors only measured cell viability but did not evaluate other indicators of osteoarthritis in general when establishing an osteoarthritis model and evaluating the therapeutic effects of SHED-derived exosomes and hyaluronic acid. It is recommended that the authors add relevant experiments.

5. The authors mentioned in their results that exosomes inhibit TNF-α and IL-6, but there was no statistical difference between the exosome-treated group and the IL-1β-treated group in Fig 3B, D.

6. The authors needed to perform exosome uptake experiments on SW1353 cells after extracting exosomes.

7. In the results section 3.3.3 of the article, the authors claim that "Inhibition of COX-2 did reduce the induction of PGE2", which is not supported by sufficient data. This is highly speculative and the analyses of hormone. The authors should remove these claims or substantiate them with functional data.

8. Add a "limitation of the study" section. Are there any limitations or strengths? Improve this part.

9. The description of the experimental results section of the article needs to be more clearly and concisely described. For example, results 3.3 and 3.4, it is recommended that the authors improve this section.

Author Response

Response to Reviewers

The authors would like to express my sincerely thanks to the editor and reviewers for the professional comments and constructive suggests. The article was revised following the suggestions. The modifications are listed as below.

Comments and Suggestions for Authors

Reviewer1

In this study, exosomes secreted by stem cells from human exfoliated deciduous teeth (SHED) and hyaluronic acid were used to evaluate the therapeutic ability to slow down OA. SHED exosomes were first collected and characterized, and SW1353 cells were stimulated with IL-1β to produce the inflammatory features of OA and then treated with exosomes and hyaluronic acid. Osteoarthritis-related indicators and components of the extracellular matrix were examined, and exosomes were found to have an inhibitory capacity for OA no less than hyaluronic acid. But the paper needs some improvements before acceptance for publication.

  1. "ECM" appears in the first paragraph of “introduction” first. The abbreviations should be defined at the first appearance. This should be revised on other abbreviations, too.

⨠ We appreciate reviewer’s comment and revised the article accordingly.

  1. lt's partly unclear what is the purpose of this paper. Improve this part in the introduction section.

⨠ According to reviewer’s comment, we added more explanation about the purpose of this study in the last paragraph of introduction part.

  1. The clarity of some figures in the article is low, especially Fig. 1.

⨠ We appreciate reviewer’s comment and increased the image resolution for better images quality.

  1. The authors only measured cell viability but did not evaluate other indicators of osteoarthritis in general when establishing an osteoarthritis model and evaluating the therapeutic effects of SHED-derived exosomes and hyaluronic acid. It is recommended that the authors add relevant experiments.

⨠ We appreciate reviewer’s comments. In this study, we would like to provide a proof of concept to reveal the effect of exosomes from SHED and how it was involved in the inflammatory signaling pathways by the in vitro OA model. Although we would like to examine more indicators to evaluate the therapeutic effects of SHED-derived exosomes and hyaluronic acid, however, the limitation of materials, especially the shortage of SHED hindered us to conduct the advanced experiments at this stage. We hope that reviewer could understand our predicament. If funding and time permit, the reviewers' suggestions will be used in future research for more in-depth measurement and verification.

  1. The authors mentioned in their results that exosomes inhibit TNF-α and IL-6, but there was no statistical difference between the exosome-treated group and the IL-1β-treated group in Fig 3B, D.

⨠ We appreciate reviewer’s comments and revised the manuscript accordingly. Please refer to the manuscript 3.3.1 with yellow highlight.  

  1. The authors needed to perform exosome uptake experiments on SW1353 cells after extracting exosomes.

⨠ We sincerely appreciate reviewer’s comments. Although we would like to do perform the exosome uptake experiments to show the entrance of exosomes into the SW1353 cells, due to the limitation of materials, especially the shortage of SHED, we cannot conduct advanced experiments at this stage. We hope reviewer could understand our predicament. If funding and time permit, the reviewers' suggestions will be used in future research for more in-depth measurement and verification.

  1. In the results section 3.3.3 of the article, the authors claim that"Inhibition of COX-2 did reduce the induction of PGE2", which is not supported by sufficient data.This is highly speculative and the analyses of hormone. The authors should remove these claims or substantiate them with functional data.

⨠ We appreciate and agree with reviewer’s comments. We did not have sufficient data to support that, therefore, we delete the claim in the article.

  1. Add a "limitation of the study" section. Are there any limitations or strengths? Improve this part.

⨠ According to reviewer’s comment, we added a part of “limitation of the study” in the article. Please refer to the part after the discussion.

  1. The description of the experimental results section of the article needs to be more clearly and concisely described. For example, results 3.3 and 3.4, it is recommended that the authors improve this section.

⨠ We appreciate reviewer’s comments. The manuscript was revised accordingly. Please refer to the part of results 3.3 and 3.4 with yellow highlight.

Reviewer 2 Report

Comments and Suggestions for Authors

Paper entiteled „ The exosomes of stem cells from human exfoliated deciduous teeth suppress inflammation in osteoarthritis” address interesting topic of use of stromal cells derived extracellular vesicles as a potential treatment of osteoarthritis. As in general this is very interesting approach I would be cautious with strong statement about therapeutic potential of something in context of OA.

1.     Authors refers that hyaluronic acid is currently treatment option for OA. I would be very cautious of this. At this point according to American College of Rheumatology and the European League Against Rheumatism the major OA treatment are still changes in life style (decreasing patient weight, increasing physical fitness) or joint replacement. A the same time treatment in clinical practice is still focused on pain therapy – what is unfortunately associated with side effects (toxicity of drugs in long-term perspective) – this although open field for novel therapies – but it must be underlined that the road to clinical success is still very long (doi: 10.1016/S0140-6736(19)30417-9 , DOI: 10.1038/nrdp.2016.72 , DOI: 10.7326/0003-4819-157-3-201208070-00473 )

2.     I would be also extremely careful while stating that (stem) cell therapies are beneficial for OA (i.e. doi.org/10.1038/s41591-023-02632-w are showing lack of superiority of stromal-cell based therapeutic approach for OA. The key point is not to provide patients hope which cannot be fulfilled yet – but underline that research is ongoing

3.     Authors change a scale on size distribution plot of exosomes to make it more viseable (1D)?

4.     In general authros should show all points on bar plots – it will be more informative about experiments results for readers

5.     Sometimes text is hard to read – I recommend authors proofreading on professional English editing (i.e. According to Figure 2A, we know that exosomes do not have toxicity to SW1353, and the survival rate of SW1353 is higher as the concentration of exosomes increases. It also needs to test the optimal concentration of the use of inflammatory drug)

6.     and 80 μg/mL for two days. the result as shown in Figure 2A” new sentence is not starting from capital. Please revise carefully yours manuscript.

7.     How authors interpret that exosomes alone increase viability of cells? Authros use MTT asay which is providing info about metabolic activity, maybe what authors observe is encahcned metabolic activity of cells or proliferation? Please consider it and discuss carefully

8.     Authors states that exosomes are more efficient in dampening inflammation than HA – I would suggest to be less rigorous in providing such statement (i.e. for TNF-alfa and IL-6 HA works more efficiently while exosomes did not cause significant effect)

9.     Authors address topic of extracellular matrix – what is crucial for OA progression. It would be beneficial to discuss this issue I the context of physicological mechanosensitivity of stromal cells in the context of sekeltal system regenereraton (doi.org/10.18388/abp.2019_2893) as well as stromal cells from dental pulp physiology (doi.org/10.3390/cells13050375) . That’s crucial to answer question – are dental pulp derived cell a proper cells source for join disease treatment ?

10.  Also in reference to previous point – since authors address to some extend mechanobiology – related processes it would be valuable to consider impact of exosomes on at least actin cytoskeleton organization (or if possible another cellular features)

In general paper is interesting but I would suggest to shift narration to more realistic as well as address to some mechanobiological points. English editing will be crucial for making this work more attractive for readers as well. 

Comments on the Quality of English Language

I recommend profreading or even academic English editing. 

Author Response

Response to Reviewers

The authors would like to express my sincerely thanks to the editor and reviewers for the professional comments and constructive suggests. The article was revised following the suggestions. The modifications are listed as below.

Comments and Suggestions for Authors

Reviewer2

Paper entiteled „ The exosomes of stem cells from human exfoliated deciduous teeth suppress inflammation in osteoarthritis” address interesting topic of use of stromal cells derived extracellular vesicles as a potential treatment of osteoarthritis. As in general this is very interesting approach I would be cautious with strong statement about therapeutic potential of something in context of OA.

  1. Authors refers that hyaluronic acid is currently treatment option for OA. I would be very cautious of this. At this point according to American College of Rheumatology and the European League Against Rheumatism the major OA treatment are still changes in life style (decreasing patient weight, increasing physical fitness) or joint replacement. A the same time treatment in clinical practice is still focused on pain therapy – what is unfortunately associated with side effects (toxicity of drugs in long-term perspective) – this although open field for novel therapies – but it must be underlined that the road to clinical success is still very long (doi: 10.1016/S0140-6736(19)30417-9, DOI: 10.1038/nrdp.2016.72 , DOI: 10.7326/0003-4819-157-3-201208070-00473 )

⨠ We appreciate reviewer’s comments. The references recommended by reviewer were included in the introduction part, and the manuscript was revised accordingly. Please refer to the Introduction part with highlight.

  1. I would be also extremely careful while stating that (stem) cell therapies are beneficial for OA (i.e. doi.org/10.1038/s41591-023-02632-ware showing lack of superiority of stromal-cell based therapeutic approach for OA. The key point is not to provide patients hope which cannot be fulfilled yet – but underline that research is ongoing

⨠ We appreciate reviewer’s comments. The references were included in the introduction part. Please refer to the Introduction part with highlight.

  1. Authors change a scale on size distribution plot of exosomes to make it more viseable (1D)?

⨠ According to reviewer’s comment, we modified the quality of Figure 1D. please refer to modified figure.

  1. In general authros should show all points on bar plots – it will be more informative about experiments results for readers

⨠ We appreciate reviewer’s comment. Here we showed the bar plots with error bar. Although the data points were not shown on bar plots, the error bar can illustrate the variation of statistical analysis.

  1. Sometimes text is hard to read – I recommend authors proofreading on professional English editing (i.e. According to Figure 2A, we know that exosomes do not have toxicity to SW1353, and the survival rate of SW1353 is higher as the concentration of exosomes increases. It also needs to test the optimal concentration of the use of inflammatory drug)

⨠ We appreciate reviewer’s comments. The concentration of inflammatory drug (IL-1β) was determined by the result showed in Figure 2B. We added more explanation for that, please refer to the yellow highlight of 3.2.2.  

  1. “and 80 μg/mL for two days. the result as shown in Figure 2A” new sentence is not starting from capital. Please revise carefully yours manuscript.

⨠ We appreciate reviewer’s comments. The manuscript was modified and corrected according to reviewer’s comment.

  1. How authors interpret that exosomes alone increase viability of cells? Authros use MTT asay which is providing info about metabolic activity, maybe what authors observe is encahcned metabolic activity of cells or proliferation? Please consider it and discuss carefully

⨠ We appreciate reviewer’s comments. The manuscript was revised accordingly, please refer to the yellow highlight of discussion.

  1. Authors states that exosomes are more efficient in dampening inflammation than HA – I would suggest to be less rigorous in providing such statement (i.e. for TNF-alfa and IL-6 HA works more efficiently while exosomes did not cause significant effect)

⨠ We appreciate reviewer’s comment. The manuscript was revised and please refer to the yellow highlight of 3.3.1.

  1. Authors address topic of extracellular matrix – what is crucial for OA progression. It would be beneficial to discuss this issue in the context of physicological mechanosensitivity of stromal cells in the context of sekeltal system regenereraton (doi.org/10.18388/abp.2019_2893) as well as stromal cells from dental pulp physiology (doi.org/10.3390/cells13050375) . That’s crucial to answer question – are dental pulp derived cell a proper cells source for join disease treatment ?

⨠ We appreciate reviewer’s comment. According to the point of view, the manuscript was modified in discussion part, please refer to the discussion part with highlight.

  1. Also in reference to previous point – since authors address to some extend mechanobiology – related processes it would be valuable to consider impact of exosomes on at least actin cytoskeleton organization (or if possible another cellular features)

 â¨  We appreciate reviewer’s comment. According to the point of view, the manuscript was modified in discussion part, please refer to the discussion part with highlight.

Round 2

Reviewer 1 Report

Comments and Suggestions for Authors

The authors' responses in the revised manuscript addressed those critiques adequately. The resulting manuscript is much improved and can be accepted on the current version. 

Author Response

We appreciate the reviewers' professional evaluation and constructive suggestions, as well as their positive feedback on the revised manuscript.

Reviewer 2 Report

Comments and Suggestions for Authors

The authors revised the manuscript addressing almost all issues underlined in my revision. What is most important for the context of their research they provide limitations of the study as well as underline challenges in OA treatment.

This is a very important responsibility of researchers addressing use of stem cells and cell-based therapies to underline that we need to be very cautious and rigorous while talking about use of new cell-based approaches in therapy. Thus it is now strong point of this paper.

The only one point authors did not address is point nr 4 "

  1. In general authros should show all points on bar plots – it will be more informative about experiments results for readers"    

 Althought authors states that error bars indicate information about data variability they do not provide information about distribution. Nevertheless, I let this point to the Editor and author's decision - if it need to be changed. 

Author Response

The only one point authors did not address is point nr 4 "

In general authros should show all points on bar plots – it will be more informative about experiments results for readers"   

 Althought authors states that error bars indicate information about data variability they do not provide information about distribution. Nevertheless, I let this point to the Editor and author's decision - if it need to be changed.

>> We appreciate reviewer’s suggestion about the bar plot. We showed the bar plots with error bar in our results. Although the data points were not shown on bar plots, the error bar can illustrate the variation of statistical analysis. All the data process followed the standard rule of statistical analysis, the sample size and significance were described in details in every figure legends. Unfortunately, we don’t have suitable software to demonstrate the bar graph with data points as reviewer’s request. We hope reviewer could understand what hindered us. If reviewer request, we would like to provide the Excel raw data to journal editor for verifying the accuracy of our data.